# Bioaccessibility of Bioactive Compounds of ‘Fresh Garlic’ and ‘Black Garlic’ through In Vitro Gastrointestinal Digestion

**DOI:** 10.3390/foods9111582

**Published:** 2020-10-31

**Authors:** Alicia Moreno-Ortega, Gema Pereira-Caro, José Luis Ordóñez, Rafael Moreno-Rojas, Víctor Ortíz-Somovilla, José Manuel Moreno-Rojas

**Affiliations:** 1Department of Food Science and Health, Andalusian Institute of Agricultural and Fisheries Research and Training (IFAPA), Alameda del Obispo, Avda. Menéndez-Pidal, 14004 Córdoba, Spain; t22moora@uco.es (A.M.-O.); mariag.pereira@juntadeandalucia.es (G.P.-C.); josel.ordonez@juntadeandalucia.es (J.L.O.); victor.ortiz@juntadeandalucia.es (V.O.-S.); 2Departamento de Bromatología y Tecnología de los Alimentos, Campus Rabanales, Ed. Darwin-anexo Universidad de Córdoba, 14071 Córdoba, Spain; Rafael.moreno@uco.es

**Keywords:** fresh garlic, black garlic, polyphenols, organosulfur compounds, simulated in vitro digestion, bioaccessibility

## Abstract

Numerous studies have reported health benefits associated with the consumption of fresh and black garlic, which are characterized by the presence of polyphenols and organosulfur compounds (OS). This study aims to analyze the bioaccessibility of the bioactive compounds in fresh and black garlic after in vitro gastrointestinal digestion by monitoring the individual profile of these compounds by ultra-high-performance liquid chromatography coupled to high-resolution mass spectrometry (UHPLC-HRMS). Polyphenols decreased from the beginning of the digestive process, is mainly affected during intestinal digestion. Regarding the OS, the S-alk(en)yl-L-cysteine (SACs) derivatives were more influenced by the acidic conditions of the gastric digestion, while the γ-glutamyl-S-alk(en)yl-L-cysteine (GSAk) derivatives were more susceptible to intestinal digestion conditions in both the fresh and black garlic samples. In conclusion, after in vitro gastrointestinal digestion, the compounds with the highest bioaccessibility were vanillic acid (69%), caffeic acid (52%), γ−glutamyl-S-methyl-L-cysteine sulfoxide (GSMCS) (77%), and S-allylmercapto-L-cysteine (SAMC) (329%) in fresh garlic. Meanwhile, in black garlic, the main bioaccessible compounds were caffeic acid (65%), GSMCS (89%), methionine sulfoxide (262%), trans-S-(1-propenyl)-L-cysteine (151%), and SAMC (106%). The treatment (heating + humidity) to obtain black garlic exerted a positive effect on the bioaccessibility of OS compounds, 55.3% of them remaining available in black garlic, but only 15% in fresh garlic. Polyphenols showed different behavior regarding bioaccessibility.

## 1. Introduction

Garlic (*Allium sativum* L.) belongs to the *Allium* genus, which is one of the main bulbs consumed either fresh or cooked, and traditionally used as a medicinal plant around the world, or as a basic ingredient in the Mediterranean diet [1,2]. The health benefits of garlic consumption include the prevention of certain non-communicable diseases comprising some types of cancer, cardiovascular diseases, and diabetes, among others [3,4,5]. These benefits have been associated mostly with the presence of bioactive compounds, such as phenolic and organosulfur compounds [6]. For instance, the regular consumption of garlic has been linked with a reduction in the risk of some types of cancer (gastric, colorectal, lung, or breast cancer) and with beneficial effects on metabolic syndrome, by delaying lipid absorption and inhibiting cholesterol synthesis [7], and on cardiovascular diseases and hypertension, by decreasing blood pressure and oxidative stress [8]. Preventive effects on obesity by improving insulin sensitivity and decreasing adiposity have also been reported in animal models [9]. The last decade has seen an increase in the popularity of black garlic, a derived product obtained from fresh garlic under specific temperature and humidity conditions [10,11]. Although obtained from fresh garlic, its composition is quite different. It presents higher contents than fresh garlic of reducing sugars, organic acids, and bioactive compounds, particularly S-allyl-cysteine (SAC) and coumaric acid [12,13]. This results in a product with different organoleptic characteristics, such as a darker color, sweeter taste, softer texture and less pungent flavor, making it much more attractive to consumers. Despite the differences between both of them, the consumption of black garlic has been shown to improve human health [14,15,16,17], although the mechanism of how phenolic compounds or organosulfur compounds from garlic achieve this is not fully understood.

To better understand the physiological response of black garlic consumption and to compare it with fresh garlic, it is of paramount importance to evaluate the effect of the digestive processes on the stability and availability for the absorption of black garlic bioactives within the human gastrointestinal tract [18]. Simulated gastrointestinal digestion is an in vitro model that aims to simulate the physiological conditions of the upper gastrointestinal tract, namely, the oral, gastric and small intestine phases, and it is frequently used to evaluate the bioaccessibility of a wide variety of bioactive compounds in foods [19,20]. For instance, numerous studies have analyzed the bioaccessibility of phenolic compounds in different food matrices, including berries [21], plant-based beverages [22], tomato [23], and coffee [20]. However, there is limited data on the bioaccessibility of organosulfur compounds, some of the main bioactive compounds of the *Allium* genus. A recent study by Torres-Palazzolo et al. [24] reported the bioaccessibility of different organosulfur compounds in raw versus cooked garlic, showing a 100% bioaccessibility of allicin in raw garlic, while in cooked garlic the bioaccessibility presented 52, 57, 66, and 87% for ajoene, 2-vinyl-4H-1,3-dithiin (2VD), diallyl disulfide (DADS) and diallyl trisulfide (DATS), respectively. To our knowledge, there are no more studies into the effect of gastrointestinal digestion on the individual profile of the phenolic and organosulfur compounds in fresh and black garlic. Therefore, the aim of this study was to determine the impact of simulated gastrointestinal digestion on both fresh and black garlic by determining the bioaccessibility of the (poly)phenols and organosulfur compounds by ultra-high-performance liquid chromatography coupled to high-resolution mass spectrometry (UHPLC-HRMS) and to evaluate the effect of the treatment used in black garlic elaboration on the bioaccessibility of bioactive compounds.

## 2. Materials and Methods

### 2.1. Chemicals

For simulated salivary, gastric, and intestinal fluids (SSF, SGF, and SIF), sodium chloride and magnesium chloride hexahydrate were purchased from Fisher Scientific (Madrid, Spain); sodium bicarbonate and ammonium carbonate were supplied by Sigma-Aldrich (Madrid, Spain); and potassium dihydrogen phosphate was obtained from VWR International Eurolab (Barcelona, Spain). The following were used for digestion: α-amylase from human saliva (300–1500 U/mg protein), pepsin (3.2–4.5 U/mg protein), and pancreatin from porcine pancreas (4 × UPS). Bile salts and calcium chloride were purchased from Sigma-Aldrich (Madrid, Spain), HCl was obtained from Merck (Darmstadt, Germany), and NaOH was acquired from Fisher Scientific (Madrid, Spain). Reference flavonoid compounds, including gallic, caffeic, coumaric, and ferulic acids, catechin, and (−)-epicatechin were purchased from Sigma-Aldrich (Madrid, Spain). Alliin, s-allyl-L-cysteine (SAC), and formic acid (FA) were acquired from Sigma-Aldrich (Madrid, Spain). Ammonium acetate, ammonium formate, and ethanol were obtained from Sigma-Aldrich. Acetonitrile and methanol were of LC-MS grade.

### 2.2. Materials and Sample Preparation

Fresh and black garlic (Spring White Garlic) were supplied by a local supplier (La Abuela Carmen^®^, Córdoba, Spain). The black garlic samples were obtained from fresh garlic using the protocol previously described [25]. The fresh and black garlic were peeled, and their cloves were ground to a final particle size of 10 µm using a cryogenic grinder with liquid nitrogen mill equipment (Freezer Mill model 6870, Fisher Scientific, Waltham, MA USA) and stored at −80 °C until the in vitro gastrointestinal digestion process.

### 2.3. In Vitro Gastrointestinal Digestion

The in vitro gastrointestinal digestion protocol previously applied by Juániz et al. [26] was adapted to our laboratory. Briefly, the whole process took place in a stirred water bath (Unitronic Reciprocating Shaking Bath model 6,032,011, J.P. Selecta, Barcelona, Spain) at 37 °C with 100 mL amber glass bottles containing 2 g of each sample in triplicate. For the development of the oral phase, 14 mL of SSF solution (Appendix A) was added to the bottles with the samples, together with 250 μL of α-amylase solution (1.3 mg/mL), 0.1 mL of 0.3 M CaCl2, and 5.65 mL of distilled water. The mixture was shaken at 37 °C for 30 min. Then, the gastric phase was started by adjusting the mixture to pH 3 using a 1 M HCl solution. The following step involved adding 15 mL of SGF solution (Appendix A) to the samples, together with 1.19 mL of a pepsin solution (100 mg of pepsin/1 mL of 0.1 M HCl solution), 0.01 mL of 0.3 M CaCl2, and 3.8 mL of distilled water. The mixture was incubated again at 37 °C for 120 min. Next, for the intestinal phase, 22 mL of SIF solution was added to the samples (Appendix A) together with 10 mL of pancreatin solution (8 mg/mL), 5 mL of bile salts (25 mg/mL), 0.08 mL of 0.3 M CaCl2, and 9.92 mL distilled water. Then, 1 M NaOH solution was employed to adjust the pH to 7. Finally, the mixture was incubated for 120 min at 37 °C.

Samples were taken before before oral digestion (BOD) and after oral digestion (AOD) and after gastric (AGD) and intestinal digestion (AID). These samples were lyophilized and stored at −80 °C until extraction and analysis.

### 2.4. Phenolic and Organosulfur Compounds Extraction and Analysis

A previous optimized and validated methodology was used for the extraction and analysis of the organosulfur and phenolic compounds from the digested fresh and black garlic samples [27]. Briefly, 0.5 of fresh or black garlic lyophilized and ground was mixed with 5 mL of deionized water-methanol (50:50, *v/v*) for 2 min at room temperature, and the mixture was sonicated for 15 min and then centrifuged at 4900 rpm for 15 min. The supernatant was collected, and residues were re-extracted twice using 5 mL of the same solvent by following the same protocol described previously. All the supernatants were pooled and frozen at −80 °C until UHPLC-HRMS analysis.

The analysis of polyphenols and organosulfur compounds in fresh and black garlic extracts were carried out using a UHPLC-PDA-MS mass spectrometer system (Thermo Scientific, San José, CA, USA) comprising of a UHPLC pump, a PDA detector scanning from 200 to 600 nm, and an autosampler operating at 4 °C (Dionex Ultimate 3000 RS, Thermo Corporation). Separation of flavonoids was performed on a 100 × 2.1 mm i.d. 1.8 μm Zorbax SB-C18 RRHD column (Agilent, Santa Clara, CA, USA) preceded by a guard pre-column of the same stationary phase and maintained at 40 °C. The mobile phases, A—acidified water 1% formic acid, and B—acetonitrile, were pumped at a flow rate of 0.15 mL min^−1^ with a 33 min gradient starting in 3% B and maintained during 1 min, then rising 60% B in 24 min, maintained during 3 min and then rising 70% B in 5 min. After that, the column was equilibrated to the previous conditions within 5 min. The separation of organosulfur compounds in fresh and black onion extracts were based on a 2.1 × 150 mm ACQUITY UPLC 1.7 μm BEH amide column (equipped with an ACQUITY UPLC BEH amide 1.7 μm van-guard pre-column) (Waters, Spain), which was maintained at 35 °C and eluted using two mobile phases: A—deionized water with 5 mM of ammonium acetate, 5 mM ammonium formate, and 1% formic acid, and B—acetonitrile, over the course of 20 min at 0.4 mL min^−1.^ The gradient started with 5% of A rising 10% A in 0.5 min, then rising 30% A in 8 min following 46% of A after 4.5 min and finally return to 5% A in 3 min and maintained during 4 min to equilibrate the column to the initial conditions.

After passing through the flow cell of the PDA detector, the column eluate went directly to an Exactive Orbitrap mass spectrometer (Thermo Scientific, San José, CA, USA) fitted with a Heated Electrospray Ionization Probe (HESI) operating in positive ionization mode for the determination of OS compounds and in negative ionization mode for the determination of polyphenols [27].

The identification of polyphenols and OS compounds were achieved as follows: (i) By comparing the exact mass and the retention time with available standards, (ii) in the absence of standards, compounds were tentatively identified by comparing the exact theoretical mass of the molecular ion with the measured accurate mass of the molecular ion and searched against metabolite databases, including Metlin, Phenol Explorer, and more general chemical databases, such as PubChem and ChemSpider. Compounds having molecular masses within the pre-specified tolerance (≤5 ppm) of the query masses are retrieved from these databases. The quantification of phenolic compounds and OS compounds was carried out by selecting the exact theoretical mass of the molecular ion by reference to standard curves prepared in diluted fresh and black garlic extracts, obtaining a linear regression analysis with *R^2^* values of >0.998 (*n* = 6). In the absence of reference compounds, they were quantified by reference to the calibration curve of a closely related parent compound. The detection and quantification limits varied from 0.0004 to 0.007 ng µL^−1^ and from 0.012 to 0.024 ng µL^−1^ for polyphenols and were 0.03 and 0.1 ng µL^−1^, for OS compounds, respectively.

### 2.5. Bioaccessibility of (Poly)Phenols and Organosulfur Compounds

The bioaccessibility index was used for calculating the percentage of bioaccessibility of (poly)phenols and organosulfur compounds after simulated gastrointestinal digestion [28,29].
Bioaccessibility = FC/IC * 100
where FC is the Final Concentration (compound concentration after simulated gastrointestinal digestion), and IC is the Initial Concentration (compound concentration before simulated gastrointestinal digestion).

### 2.6. Statistical Analysis

Statistical analyses were performed based on six replicate measures of each sample. A one-way ANOVA was carried out using R software (v. 3.6.3, R Core Team, Vienna, Austria) to determine significant differences between the stages of the in vitro gastrointestinal digestion, the significance being accepted for a *p*-value < 0.05. Next, Fisher’s LSD pairwise comparison was performed on the data.

## 3. Results and Discussion

### 3.1. Bioaccessibility of Phenolic Compounds after Simulated Gastrointestinal Digestion

Before oral digestion, a total of five (poly)phenols were identified in the fresh garlic and four in the black garlic (Table 1, Figure 1). Details of their identification and quantification are presented in the Appendix A, Appendix A. Among phenolic acids, caffeic acid and gallic acid were the main compounds in the fresh garlic, accounting for 95.3% of the total phenolic content, while these compounds plus coumaric acid were the main ones found in the black garlic, representing 99.4% of the total polyphenol content. These results are in line with those reported by Kim et al. [30], who demonstrated that caffeic acid was the major phenolic compound in fresh garlic, while in black garlic, it was coumaric acid.

After in vitro oral digestion, the total polyphenol content of the fresh garlic significantly increased (1001 nmol/g FW, Fresh Weight), accounting for a mean recovery of 185.9%. This increase was mainly attributed to the 1.5 and 1.8-fold increase in the levels of caffeic acid and gallic acid, respectively, along with the appearance of a substantial amount of benzoic acid (162 nmol/g FW) (Table 1). During the oral digestion, epicatechin, and chlorogenic acid also appeared in the fresh garlic at low concentrations (0.09 and 0.05 nmol/g FW, respectively). The increase in hydroxycinnamic acids, including gallic, coumaric, and ferulic acid, during the oral stage of the simulated digestion of soymilk [31], and flours from persimmon fruit [32] has been reported by other authors. It is considered to be mainly due to pH and enzymatic activity, which could induce the breakdown between these compounds and other food components.

After in vitro gastric digestion, the content of most of the (poly)phenols in the fresh garlic showed a notable decrease, with recoveries ranging from 55.9% for ferulic acid to 81.3% for caffeic acid (Table 1); while the concentration of gallic acid (203% recovery with regard to the initial content in the fresh garlic) remained stable compared with its concentration after oral digestion. The flavonoid epigallocatechin also showed a significant increase in its concentration after gastric digestion with a recovery of 230%, although it remained a minor component. These results agree with those reported by Lucas-González et al. [32], who found an increase in the gallic acid content of 160 and 176.7% in two types of flours obtained from persimmon fruit after oral and gastric digestion. In addition, catechin appeared for the first time during this stage, although it was a marginal compound. Despite the significant decrease in some polyphenols after gastric digestion, the total phenolic content of the fresh garlic after in vitro gastric digestion was 646 nmol/g FW, representing a mean recovery of 120% at the end of this stage (Table 1). During intestinal digestion, the last step of the digestion process, there was a significant reduction in the total content of (poly)phenols, 58.6% being the mean bioaccessibility index of (poly)phenols in the fresh garlic (Table 1). Individually, most of the polyphenols presented initially in the fresh garlic significantly decreased their concentration after in vitro digestion, vanillic acid (69.4%), and ferulic acid (51.5%) being the phenolics that presented the highest bioaccessibility, followed by epigallocatechin (35.3%). Meanwhile, gallic acid and caffeic acid were more likely to be affected by the in vitro digestion, with bioaccessibility indexes of 25.2 and 24.6%, respectively. It is worth mentioning that benzoic acid, catechin, epicatechin, and chlorogenic acid appeared after in vitro digestion, but were not identified in fresh garlic. These compounds arguably come from the breakdown of supramolecular structures that store phenolic compounds as the apparently weak bond of these phenolic acids with dietary fiber allows them to be released more easily during simulated gastrointestinal digestion [33]. Nevertheless, except for benzoic acid, these compounds were found in low amounts, accounting for 0.05% of the total phenolic content after the complete digestion.

Regarding black garlic, the content of almost all the (poly)phenols significantly decreased after oral digestion, with a mean recovery of 86.2%. Phenolics, such as gallic acid, coumaric acid, and epigallocatechin gallate, presented recoveries of 76.7, 68.2, and 79.4% respectively, while caffeic acid seemed not to be seriously affected by oral digestion, with a recovery of 92.3%.The stability of caffeic acid after oral digestion in black garlic could be a consequence of a more complex black garlic matrix, which may affect its bioaccessibility. Indeed, it has been shown that the matrix components, such as the proteins, carbohydrates, and fiber, could affect phytochemical bioaccessibility, and therefore, their release from the food matrix during gastrointestinal digestion [31,34].

After gastric digestion, the total content of phenolic compounds in black garlic significantly decreased, with a recovery of 74.9%. Two phenolic compounds, gallic acid, and epigallocatechin gallate, disappeared after this step. Likewise, the content of caffeic acid, the main phenolic compound in black garlic, slightly decreased, with a recovery of 79.6% with regard to its initial amount. However, the content of coumaric acid increased by 31.0% during this stage. Moreover, during the last in vitro digestion step, the in vitro intestinal digestion, there was a significant loss in the quantity of almost all the (poly)phenols, caffeic acid being the only (poly)phenol present after the in vitro digestion process of the black garlic, with a bioaccessibility index of 64.8% (Table 1). Overall, the total phenolic content of the black garlic during the whole in vitro digestion process decreased, the mean bioaccessibility index of (poly)phenols in black garlic being 47.2% (Table 1).

In summary, the total amount of (poly)phenols in the fresh garlic increased significantly after oral digestion, mainly due to the notable emergence of benzoic acid and the significant increase in caffeic and ferulic acid (Figure 2). Subsequently, the (poly)phenolic content decreased during the rest of the gastrointestinal digestion, this being more pronounced during the intestinal digestion. This trend could be related to the fact that the phenolic content in garlic is mainly represented by phenolic acids, which are more unstable under the conditions of intestinal digestion (basic pH), while at acid pH, these compounds are easily released from the matrix [35]. In contrast, in the black garlic, although there was a significant decrease in the content of polyphenols during the gastrointestinal digestion (Figure 2), with only caffeic acid remaining bioaccessible, the effect of the gastric and intestinal conditions was not so marked when compared with fresh garlic. Moreover, it is noteworthy that the gastrointestinal digestion had a negative effect on the bioaccessibility of the polyphenols in both matrices, 58.6% and 42.7% remaining bioaccessible in fresh garlic and black garlic, respectively.

### 3.2. Bioaccessibility of Organosulfur Compounds after Simulated Gastrointestinal Digestion

A total of 23 organosulfur compounds (OS) were identified and quantified in the fresh garlic before the simulated gastrointestinal digestion, including 13 S-alk(en)yl-L-cysteine derivatives (SACs) and 10 γ-glutamyl-S-alk(en)yl-L-cysteine derivatives (GSAks), which accounted for 76.8% and 23.1% of the total OS in the samples, respectively (Table 2, Figure 3). Details of their identification and quantification are presented in Appendix A (Appendix A). Among them, alliin (41706 nmol/g FW), cycloalliin (5251 nmol/g FW), methiin (4425 nmol/g FW), and trans-S-(1-propenyl)-L-cysteine (S1PC) (2909 nmol/g FW) were the main S-alk(en)yl-L-cysteine (SACs) derivatives in the fresh garlic, while γ−glutamyl-S-allyl-L-cysteine (GSAC) (13,714 nmol/g FW), γ-glutamyl-S-methyl-L-cysteine (GSMC) (1580 nmol/g FW) and γ-glutamyl-S-(1-propenyl) cysteine sulfoxide (G1PCS) (1311 nmol/g FW) were the main GSAk derivatives. These results are in line with those reported by Goncharov et al. [36], who found that alliin is the major organosulfur compound in raw garlic along with other organosulfur derivatives, including S1PC.

After oral digestion, the total content of organosulfur compounds in the fresh garlic decreased notably, showing a mean recovery of total OS of 44.4% (Table 2). The compounds mostly affected during the oral digestion were the SACs derivatives, with a mean recovery of 37.1%, while the GSAk derivatives were less affected, with a mean recovery of 68.6%. Among them, the levels of γ–glutamyl-S-(S-1-propenyl)cysteine-glycine, γ–glutamyl-S-(S-1-methyl)cysteine-glycine, γ–glutamyl-S-(S-1-propyl) cysteine (GS1PC), γ–glutamyl-S-allyl-L-cysteine (GSAC), γ–glutamyl-S-allylmercaptocysteine (GSAMC) and γ–glutamyl-cysteine decreased, which is probably, at least in part, a consequence of the enzyme hydrolysis during the oral phase [37] resulting in the loss of the γ–glutamyl group and in the formation of other OS compounds. For instance, S-allylmercapto-L-cysteine (SAMC), with a recovery of 156% after oral digestion, likely originates from the hydrolysis of GSAMC, as has been indicated in Figure 4.

After the gastric digestion of the fresh garlic samples, a reduction was observed in the concentration of almost all the OS compounds, with a mean recovery of 27.2%, highlighting the susceptibility of these compounds to simulated gastric conditions. The mean recovery of GSAk derivatives was 44.6%, ranging from 21.3% for γ–glutamyl-S-(S-methyl) cysteine-glycine to 76.1% for GSMCS; while for SACs derivatives, the recovery ranged from 7.2% for propiin to 62.2% for NASAC, with a mean recovery of 22.0%. It is of note that the concentration of SAMC continued to increase after gastric digestion, accounting for 181.1% of recovery (Table 2). This could be explained by the fact that SAMC is a metabolite from alliin produced when it reacts with cysteine after ingestion of fresh garlic [38]. This is arguably due to the hydrolysis of its precursor GSAMC, whose concentration decreased by almost 90% during the in vitro digestion (Figure 4) [39].

At the end of the digestion process, the organosulfur compounds from the fresh garlic showed a total mean bioaccessibility of 15.3%, GSAK derivatives (26.3%) showing a higher bioaccessibility index than SACs derivatives (12.0%). The organosulfur compound GSAC, followed by GSMC and G1PCS, was the main GSAk derivative observed after the gastrointestinal digestion of the fresh garlic (38.7%), while alliin, methiin, cycloalliin, SAC, SAMC, and S1PC were the main SACs derivatives, accounting for 56.8% of the total OS content. The OS compounds with the highest bioaccessibility indexes were SAC, methionine sulfoxide, GSMCS, and SAMC, with 60.8, 72.4, 76.6, and 329.3%, respectively. To the best of our knowledge, this is the first time that the in vitro gastrointestinal stability of fresh garlic OS has been studied.

Regarding the black garlic, a total of 17 organosulfur compounds were identified and quantified (Table 2, Figure 3), of which 64.1% were SACs derivatives, SAC (30.5%), and alliin (22.3%) being the main ones. The main GSAk derivatives were GSAC (17.4%) and G1PCS (13.7%), accounting for 31.1% of the total OS in the black garlic (Table 2). These data are in line with Molina-Calle et al. [40], who also tentatively identified SAC, alliin, and GSAC in the extracts from black garlic.

After the oral digestion of the black garlic, OS compounds showed better stability than in the fresh garlic, with an average of 88.3%, the recovery rates ranging from 80.5 to 106.6% (Table 2). The significant 1.5-fold increase in S-(S-propyl) cysteine after the oral digestion of the black garlic is noteworthy (Table 2). After gastric digestion, significant changes were observed in the concentration of GSAC and G1PCS, which lead to a significant decrease in the concentration of GSAk derivatives after this digestion step with a mean recovery of 82.5%. Moreover, some SACs derivatives, including S-allylglutathione and S-(S-propyl) cysteine disappeared, and others, such as SAC, SAMC, the lacrimatory factor, alliin, and methionine sulfoxide were strongly affected by the gastric conditions, their concentrations decreasing significantly; other SACs derivatives, such as deoxymethiin, S-(2-carboxypropyl)cysteine, S1PC, and methiin, were stable or their concentrations even significantly increased (Table 2).

After intestinal digestion, the total OS content in the black garlic decreased significantly, with a mean recovery of 55.3%. Particularly, GSAk derivatives were seriously affected, showing a mean recovery of 42.7%. This marked a decrease in their concentration is likely due to the great instability of these compounds under alkaline pH conditions (intestinal digestion), consistent with earlier reports [41]. Moreover, compounds, such as methiin, deoxymethiin, S-(2-carboxypropyl) cysteine, GSMC, and γ–Glutamyl-S-(S-1-propenyl) cysteine-glycine, were not detected after the intestinal phase. Conversely, three organosulfur compounds, namely, SAMC, S1PC, and methionine sulfoxide, showed an increase in their concentration during the simulated intestinal digestion, with respective recovery rates of 106.4, 151.2, and 262.0%. The 3-fold increase in methionine sulfoxide after gastrointestinal digestion is probably due to the oxidation of methionine during the in vitro digestion [42].

Overall, the bioaccessibility index of total organosulfur compounds was higher for black than fresh garlic, 55.3 and 15.3%, respectively (Figure 5). Indeed, 7 of the 13 organosulfur compounds found in both matrices—GSMCS, G1PCS, GSAMC, alliin, methionine sulfoxide, trans-S-(1-propenyl)-L-cysteine and lacrimatory factor— showed higher bioaccessibility indexes in black garlic (Table 2). Regarding the individual OS compounds, alliin was the main SACs derivative present in both matrices, presenting a bioaccessibility of 5.3 and 76.6% in fresh and black garlic, respectively. This means that even though the content of alliin in fresh garlic was 7-fold higher than in black garlic before oral digestion, after this process, its content was almost 2-fold higher in black garlic, suggesting a higher bioaccessibility for this compound in the black garlic. In contrast, SAMC showed a significantly higher bioaccessibility index in fresh (329.3%) compared to black garlic (106.4%), despite presenting 164 nmol/g FW and 408 nmol/g FW, respectively, before digestion. Related to this, Xiao et al. [43] and Yi et al. [44] reported that SAMC presented activity that inhibited tumor growth in in vivo models. Furthermore, as discussed above, SAMC is a metabolite derived from the metabolic pathways of GSAMC [38,39], and the decrease in this compound was evidenced throughout the digestive process. Conversely, a higher percentage of bioaccessibility of methionine sulfoxide was found in the black garlic, with 262.0%, compared with 72.6% in the fresh garlic.

Taking the results from this study together, a putative breakdown pathway of OS compounds in fresh and black garlic during in vitro gastrointestinal digestion is proposed (Figure 4). The first step represents the scission mediated by hydrolysis and gamma-glutamyl to yield the corresponding S-alk(en)yl-L-cysteine derivatives, which subsequently can be further converted by oxidation of the S-group to sulfoxide derivatives.

## 4. Conclusions

Considering the limited information available, the present study was undertaken to investigate the effects of in vitro simulated gastrointestinal digestion on the recovery and bioaccessibility of the individual (poly)phenolic and OS compounds in black garlic, and compare the results with its counterpart (fresh garlic). A notable increase in polyphenol compounds was observed during the oral digestion of fresh garlic, followed by a significant decrease during the subsequent steps of the gastrointestinal digestion, benzoic acid being the main polyphenol remaining at the end of the digestion process. Meanwhile, in black garlic, the polyphenol content decreased from the beginning of the digestive process, caffeic acid the only polyphenol still remaining. Regarding OS content, SACs derivatives were more influenced by the gastric digestion, while GSAk derivatives were more sensitive to intestinal digestion conditions in both the fresh and black garlic. Conversely, OS compounds in the black garlic presented greater stability throughout the digestive process than those in the fresh garlic. The bioaccessibility indexes of OS compounds in the fresh garlic ranged from 3.2% for propiin to 329.3% for SAMC, while in the black garlic, they ranged from 21.3 to 262% for GSAC and methionine sulfoxide, respectively. The heat treatment to obtain black garlic has a positive effect on OS bioaccessibility from garlic, but not on polyphenols. A plausible route for the breakdown of OS compounds during gastrointestinal digestion has been proposed.

## Figures and Tables

**Figure 1 foods-09-01582-f001:**
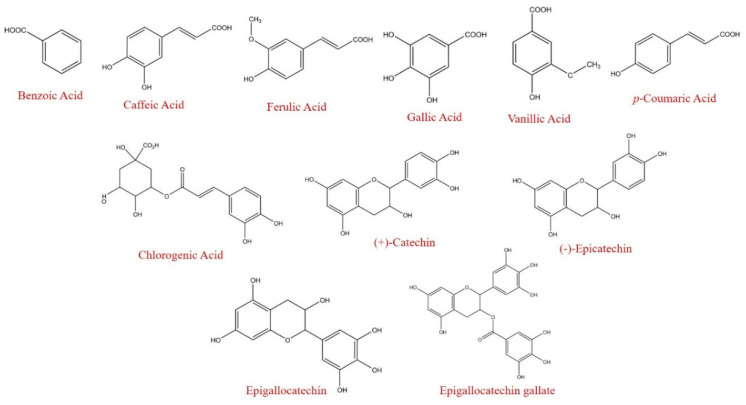
Chemical structures of fresh and black garlic polyphenols.

**Figure 2 foods-09-01582-f002:**
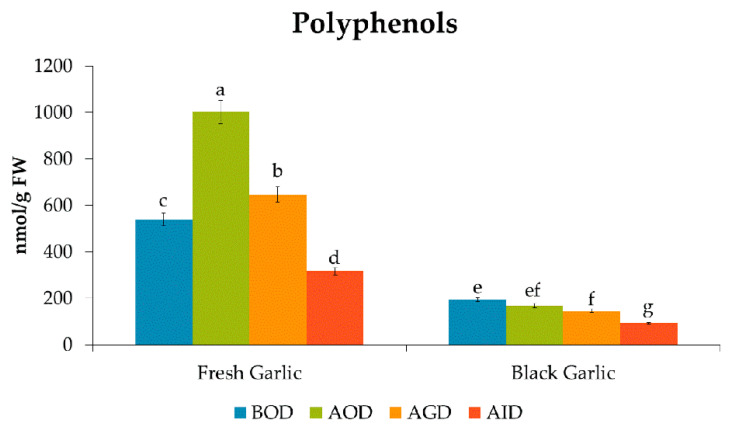
Quantities of total polyphenols during the in vitro gastrointestinal digestion. Data are expressed as nmol/g FW as mean values (*n* = 6). Different letters (one-way ANOVA) denote statistically significant differences between the stages of simulated gastrointestinal digestion (*p*-value < 0.05). (BOD, Before Oral Digestion; AOD, After Oral Digestion; AGD, After Gastric Digestion; AID, After intestinal digestion).

**Figure 3 foods-09-01582-f003:**
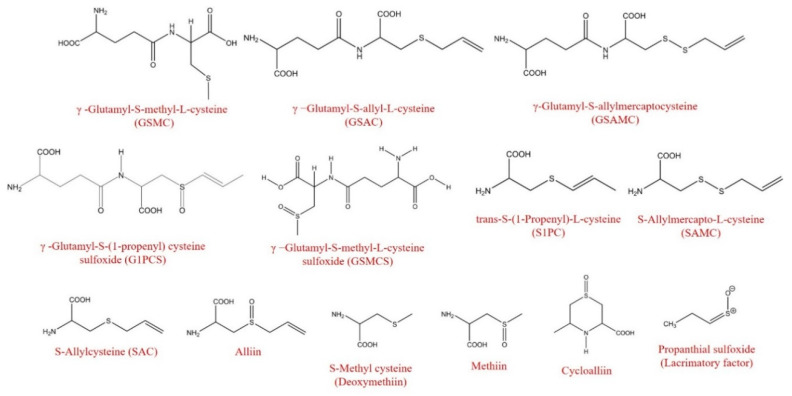
Chemical structures of fresh and black garlic organosulfur compounds.

**Figure 4 foods-09-01582-f004:**
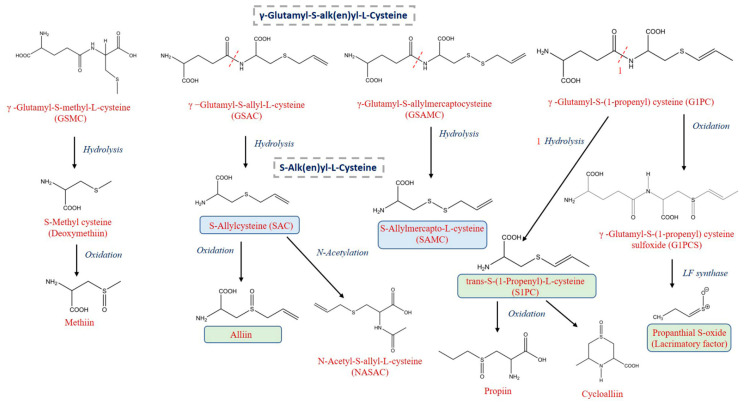
Proposed breakdown pathway of organosulfur compounds during in vitro gastrointestinal digestion.

**Figure 5 foods-09-01582-f005:**
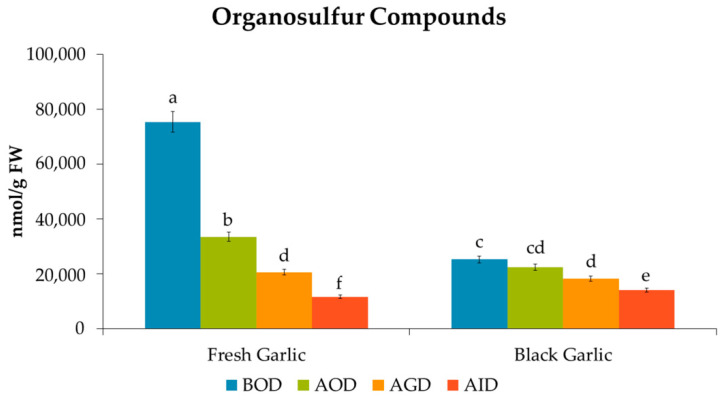
Quantities of total organosulfur compounds during the in vitro gastrointestinal digestion. Data are expressed as nmol/g FW as mean values (*n* = 6). Different letters (one-way ANOVA) denote statistically significant differences between the stages of simulated gastrointestinal digestion (*p*-value < 0.05). (BOD, Before Oral Digestion; AOD, After Oral Digestion; AGD, After Gastric Digestion; AID, After intestinal digestion).

**Table 1 foods-09-01582-t001:** Concentration (nmol/g FW) of (poly)phenols presented in fresh and black garlic samples at different stages during the simulated gastrointestinal digestion. Data are expressed as mean values (*n* = 6).

Compounds	BOD	AOD	% Recovery	AGD	% Recovery	AID	% Recovery-Bioaccessibility	*p*-Value
	*Fresh Garlic*							
Benzoic Acid	nq	162 ab	-	141 b	-	173 a	-	***
Vanillic acid	18.3 a	15.5 a	84.7	11.7 b	63.9	12.7 b	69.4	***
Gallic acid	58.7 b	105 a	180	119 a	203	14.8 c	25.2	***
Caffeic acid	455 b	712 a	156	370 b	81.3	112 c	24.6	***
Ferulic Acid	6.8 a	5.6 b	82.4	3.8 c	55.9	3.5 c	51.5	***
(+)-Catechin	nq	nq	-	0.01 b	-	0.11 a	-	***
(−)-Epicatechin	nq	0.092 a	-	0.04 b	-	0.03 c	-	***
Epigallocatechin	0.19 c	0.39 b	203	0.44 a	230.0	0.07 d	35.3	***
Chlorogenic acid	nq	0.054 a	-	0.04 b	-	0.02 c	-	***
Total	539 c	1001 a	185.9	646 b	120.0	316 d	58.6	***
	*Black Garlic*							
Gallic Acid	14.7 a	11.2 b	76.7	nq	-	nq	-	***
Caffeic Acid	142 a	131 ab	92.3	113 b	79.6	92 c	64.8	***
Coumaric Acid	37.1 a	25.3 c	68.2	33.1 b	89.4	nq	-	***
Epigallocatechin gallate	1.2 a	0.95 b	79.4	nq	-	nq	-	***
Total	195 a	168 b	86.2	146 c	74.9	92 d	47.2	***

Different letters (one-way ANOVA) denote significant differences (*p* < 0.05) among the five stages for the same compound. Ns, non-significant; *** *p*-value < 0.001. nq, No-quantified. BOD, Before Oral Digestion; AOD, After Oral Digestion; AGD, After Gastric Digestion; AID, After Intestinal Digestion.

**Table 2 foods-09-01582-t002:** Concentration (nmol/g FW) of organosulfur compounds presented in fresh and black garlic samples at different stages during the simulated gastrointestinal digestion. Data are expressed as mean values (*n* = 6).

Organosulfur Compounds.	BOD	AOD	% Recovery	AGD	% Recovery	AID	% Recovery-Bioaccessibility	*p*-Value
	*Fresh Garlic*							
***γ-Glutamyl-S-Alk(en)yl-L-Cysteine Derivatives (GSAk)***								
γ-Glutamyl-S-(2-carboxypropyl) cysteine-glycine	287 a	253 a	88.2	163 b	56.9	71 c	24.6	***
γ-Glutamyl-S-(S-1-propenyl) cysteine-glycine	159.0 a	65.6 b	41.3	45.5 c	28.6	11.3 d	7.1	***
γ-Glutamyl-S-(S-methyl) cysteine-glycine	10.3 a	4.0 b	39.0	2.2 c	21.3	0.8 d	8.0	***
γ-Glutamyl-S-methyl-L-cysteine (GSMC)	1580 a	1490 a	94.3	899 b	56.9	598 c	37.9	***
γ-Glutamyl-S-(propenyl) cysteine (GS1PC)	6.7 a	3.5 b	52.3	2.7 b	40.3	1.5 c	22.8	***
γ-Glutamyl-S-allyl-L-cysteine (GSAC)	13,714 a	8837 b	64.4	5849 c	42.7	3328 d	24.3	***
γ-Glutamyl-S-allylmercaptocysteine (GSAMC)	343 a	158 b	46.0	97 c	28.1	35 d	10.2	***
γ-Glutamyl-cysteine	36.1 a	19.3 b	53.5	10.6 c	29.3	4.4 d	12.2	***
γ-Glutamyl-S-methyl-L-cysteine sulfoxide (GSMCS)	17.0 a	14.5 b	85.3	13.0 b	76.1	13.0 b	76.6	***
γ-Glutamyl-S-(1-propenyl) cysteine sulfoxide (G1PCS)	1311 a	1142 a	87.1	706 b	53.8	526 b	40.1	***
Total GSAk Derivatives	17,464 a	11,987 b	68.6	7788 c	44.6	4589 d	26.3	***
*S-Alk(en)yl-L-Cysteine Derivatives (SACs)*								
S-Methylcysteine (Deoxymethiin)	534 a	270 b	50.7	118 c	22.1	52 d	9.8	***
S-Allylcysteine (SAC)	1144 a	1017 a	88.9	384 c	33.6	696 b	60.8	***
S-(2-Carboxypropyl) cysteine	88 a	50 b	57.0	34 c	38.4	21 d	24.1	***
S-allylglutathione (SAG)	2.7 a	1.4 b	52.9	0.8 c	29.2	0.4 d	14.2	***
trans-S-(1-Propenyl)-L-cysteine (S1PC)	2909 a	1868 b	64.2	1216 c	41.8	662 d	22.8	***
S-Allylmercapto-L-cysteine (SAMC)	164 c	256 b	156.4	296 b	181.1	539 a	329.3	***
S-Allylsulfenic acid (Lacrimatory factor)	1517 a	887 b	58.5	624 c	41.2	291 d	19.2	***
Alliin	41,706 a	11,621 b	27.9	5937 c	14.2	2207 c	5.3	***
S-Methyl-l-cysteine sulfoxide (Methiin)	4425 a	2651 b	59.9	2176 c	49.2	1668 d	37.7	***
S-Propyl-L-cysteine sulfoxide (Propiin)	26.89 a	3.7 b	13.7	1.9 bc	7.2	0.9 c	3.2	***
Cycloalliin	5251 a	2741 b	52.2	1855 c	35.3	771 d	14.7	***
Methionine sulfoxide	14.5 a	8.4 c	57.9	8.7 c	60.0	10.5 b	72.4	***
N-Acetyl-S-allyl-L-cysteine (NASAC)	60.3 b	69.1 a	114.7	37.5 c	62.2	10.4 d	17.2	***
Total SACs Derivatives	57,842 a	21,444	37.1	12,689 c	22.0	6929 d	12.0	***
Total OS Compounds	75,306 a	33,431 b	44.4	20,477 c	27.2	11,518 d	15.3	***
	*Black Garlic*							
***γ-Glutamyl-S-Alk(en)yl-L-Cysteine (GSAk)***								
γ-Glutamyl-S-(S-1-propenyl) cysteine-glycine	119 a	120 a	100.9	120 a	101.1	nd	-	***
γ-Glutamyl-S-methyl-L-cysteine (GSMC)	199 a	206 a	103.7	200 a	100.7	nd	-	***
γ-Glutamyl-S-allyl-L-cysteine (GSAC)	4393 a	3900 ab	88.8	3597 b	81.9	934 c	21.3	***
γ-Glutamyl-S-allylmercaptocysteine (GSAMC)	504 a	510 a	101.2	560 a	111.3	287 b	57.1	***
γ-Glutamyl-S-methyl-L-cysteine sulfoxide (GSMCS)	380	384	101.1	340	89.4	338	88.9	ns
γ-Glutamyl-S-(1-propenyl) cysteine sulfoxide (G1PCS)	3452 a	3346 a	96.9	2645 b	76.6	2305 b	66.8	***
Total GSAk Derivatives	9046 a	8466 a	93.6	7463 b	82.5	3864 c	42.7	***
*S-Alk(en)yl-L-Cysteine (SACs)*								
S-methyl cysteine (Deoxymethiin)	168 b	166 b	98.7	218 a	129.8	nd	-	***
S-Allyl-L-cysteine (SAC)	7683 a	6189 b	80.5	4066 c	52.9	2803 d	36.5	***
S-(2-carboxypropyl) cysteine	115 b	116 b	100.7	149 a	130.3	nd	-	***
S-allylmercapto-L-cysteine (SAMC)	408 ab	358 b	87.7	224 c	54.9	434 a	106.4	***
S-allylglutathione	46.6 a	43.6 a	93.6	nd	-	nd	-	
trans-S-(1-Propenyl)-L-cysteine (S1PC)	205 c	198 c	96.6	243 b	118.5	310 a	151.2	***
S-allylsulfenic acid (Lacrimatory factor)	935 a	818 ab	87.5	648 b	69.4	895 a	95.7	**
S-(S-propyl) cysteine	229 b	355 a	155.3	nd	-	nd	-	***
Alliin	5611 a	4735 ab	84.4	4423 b	78.8	4302 b	76.7	**
S-methyl-cysteine sulfoxide (Methiin)	243 a	255 a	104.9	235 a	96.6	nd	-	***
Methionine sulfoxide	502 b	535 b	106.6	411 c	81.9	1314 a	262.0	***
Total SACs Derivatives	16,145 a	13,768 b	85.3	10,617 c	65.8	10,059 c	62.3	***
Total OS Compounds	25,191 a	22,234 b	88.3	18,080 c	71.8	13,923 d	55.3	***

Different letters (one-way ANOVA) denote significant differences (*p* < 0.05) among the five stages for the same compound. Ns, non-significant; ** *p*-value < 0.01; *** *p*-value < 0.001. nd, No-detected. BOD, Before Oral Digestion; AOD, After Oral Digestion; AGD, After Gastric Digestion; AID, After Intestinal Digestion.

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
