# Peer review of "Bioaccessibility of Bioactive Compounds of ‘Fresh Garlic’ and ‘Black Garlic’ through In Vitro Gastrointestinal Digestion"

_foods, 2020, doi:10.3390/foods9111582_

Round 1

Reviewer 1 Report

Minor revision

This is a standard article about the bioaccessibility of bioactive compounds from foods.

Overall English is good and the manuscript is clear. 

Allium should be written in italic and the initial letter of the botanist is missing. I would be grateful if the authors may correct the botanical name of the plants reported through the manuscript according to the botanical rules.

I would be grateful if the authors may uniform ‘in vitro’ in italic through the manuscript.

Introduction: I would be grateful if the authors may improve the literature regarding the mediterranean diet with these references

1. D. Metro, R. Tardugno, M. Papa, C. Bisignano, L. Manasseri, G. Calabrese, T. Gervasi, G. Dugo, N. Cicero (2017). Adherence to the Mediterranean diet in a Sicilian student population. Natural Product Research, DOI: 10.1080/14786419.2017.1402317.

2. D. Metro, M. Papa, L. Manasseri, T. Gervasi, L. Campone, V. Pellizzeri, R. Tardugno*, G. Dugo (2018). Mediterranean diet in a Sicilian student population. Second part: breakfast and its nutritional profile. Natural Product Research, DOI: 10.1080/14786419.2018.1452016.

Materials and Methods: I would be grateful if the authors may add the  chromatographic data, the instrument model, compound names, linearity data such as Lod, Loq and r2 values of the calibration curves. 

Results and discussion: clear.

Conclusion: clear.

Reviewer 2 Report

Moreno-Ortega et al. report on the bioaccessibility of bioactive compounds from fresh garli and black garlic using a simulated model of human gastrointestinal digestion. The topic is interesting and the methodology is sound. It would be interesting to evaluate the effect of a food matrix on the bioaccessibility of these compounds. The effect of milling could also be discussed. 

Reviewer 3 Report

The manuscript showed interesting investigation of polyphenols and organosulfur compounds in fresh garlic and black garlic through in vitro gastrointestinal digestion. However, there are few significant issues need to be addressed:

Page 1; Line 16: Abstract section: Abstract need to be constructed in a way that all the outcomes with the significant figures should be given either in the percentage or statistically significant values.

Page 3; Line 118: The method for Phenolic and Organosulfur Compounds Extraction and Analysis required more detailed description.

Page 6; Line 148: What are the mechanistic reasons behind the increase in the caffeic acid and gallic acid after oral digestion in fresh garlic? Describe in details.

Page 6; Line 165: What are the mechanistic reasons behind the increase in gallic acid only after gastric digestion in fresh garlic? Describe in details.

Page 7; Line 189: Why caffeic acid was almost unaffected by oral digestion in black garlic?

Round 2

Reviewer 3 Report

It was good to see the comments response for the authors, the authors addressed all the comments and there are no further comments required from my side.